# From Cohort to Cohort: A Similar Mixture Approach (SMACH) to Evaluate Exposures to a Mixture Leading to Thyroid-Mediated Neurodevelopmental Effects Using NHANES Data

**DOI:** 10.3390/toxics11040331

**Published:** 2023-03-31

**Authors:** Maria Sapounidou, Patrik L. Andersson, Michelle Leemans, Jean-Baptiste Fini, Barbara Demeneix, Joëlle Rüegg, Carl-Gustaf Bornehag, Chris Gennings

**Affiliations:** 1Department of Chemistry, Umea University, SE-901 87 Umea, Sweden; 2UMR 7221, Phyma, CNRS–Muséum National d’Histoire Naturelle, Sorbonne Université, 75005 Paris, France; 3Department of Organismal Biology, Environmental Toxicology, Uppsala University, SE-752 36 Uppsala, Sweden; 4Faculty of Health, Science and Technology, Department of Health Sciences, Karlstad University, SE-651 88 Karlstad, Sweden; 5Department of Environmental Medicine and Public Health, Icahn School of Medicine at Mount Sinai, New York, NY 10029, USA

**Keywords:** mixtures risk assessment, causal inference, endocrine-disrupting chemicals

## Abstract

Prenatal exposure to a mixture (MIX N) of eight endocrine-disrupting chemicals has been associated with language delay in children in a Swedish pregnancy cohort. A novel approach was proposed linking this epidemiological association with experimental evidence, where the effect of MIX N on thyroid hormone signaling was assessed using the *Xenopus* eleuthero-embryonic thyroid assay (XETA OECD TG248). From this experimental data, a point of departure (PoD) was derived based on OECD guidance. Our aim in the current study was to use updated toxicokinetic models to compare exposures of women of reproductive age in the US population to MIX N using a Similar Mixture Approach (SMACH). Based on our findings, 66% of women of reproductive age in the US (roughly 38 million women) had exposures sufficiently similar to MIX N. For this subset, a Similar Mixture Risk Index (SMRI_HI_) was calculated comparing their exposures to the PoD. Women with SMRI_HI_ > 1 represent 1.1 million women of reproductive age. Older women, Mexican American and other/multi race women were less likely to have high SMRI_HI_ values compared to Non-Hispanic White women. These findings indicate that a reference mixture of chemicals identified in a Swedish cohort—and tested in an experimental model for establishment of (PoDs)—is also of health relevance in a US population.

## 1. Introduction

Humans are exposed to endocrine-disrupting chemicals (EDCs) from many different sources (e.g., plasticizers, personal care products, water, air, food, and food containers) and compounds (e.g., phthalates, phenols, per-fluorinated compounds). There is growing evidence that prenatal exposures to mixtures of EDCs are associated with adverse health outcomes, including neurodevelopmental disorders [1,2]. This is particularly problematic based on the large number of EDCs in our environment that are detected in human biomonitoring studies [3] and the fact that regulatory guidelines for EDC exposure are generally one at a time, not accounting for human-relevant unintended EDC mixture exposure.

Caporale et al. [3] reported on a novel approach linking epidemiological associations from a pregnancy cohort with experimental evidence to establish a causal relationship between exposure to a mixture of EDCs with developmental neurotoxicity, partly mediated by thyroid hormone (TH) disruption. Epidemiological evidence in the Swedish Environmental, Longitudinal, Mother and child, Asthma and allergy (SELMA) study [4] established that prenatal exposure to EDCs was associated with cognitive development in children at 2.5 years of age [5]. Language delay is an early marker of adverse neurodevelopment effects later in life such as autism, cognitive function, etc. [5]. The established ‘typical’ mixture (i.e., reference mixture) from this cohort of pregnant women including 8 compounds (Table 1) (called MIX N based on the geometric mean from the biomonitoring data) was experimentally evaluated at human-relevant levels, among others in the *Xenopus* eleuthero-embryonic thyroid assay (XETA) where the TH-disrupting capacity of the mixture was investigated [3]. We focused on the TH-axis based on its essential role in brain development reported in epidemiological and experimental evidence of EDC-regulated dysregulation [3]. A point of departure (PoD) was estimated using OECD guidance values for a benchmark response. The resulting benchmark dose estimate was translated back to human concentration values using a similar mixture approach (SMACH) [3]. In the first step, the subset of the cohort with sufficiently similar exposures to the reference mixture (MIX N) was established. In the second step, a Similar Mixture Risk Index (SMRI) was estimated based on exposure concentrations relative to the PoD, where values exceeding 1 indicate exposures of concern.

The objective of this paper was to perform a SMACH analysis and identify risk in US populations based on the derived PoD from MIX N as described in Caporale et al. [3]. To do that, biomonitoring data were extracted from the US National Health and Nutrition Examination Survey (NHANES) database and translated into plasma concentrations using gestation physiologically based toxicokinetic (PBTK) modeling. With survey sampling weights, we estimated not only the percentage of women of reproductive age in the US population with sufficiently similar exposures to MIX N, but also the percent of those women that during pregnancy would have sufficiently similar MIX N exposure in levels associated with SMRI > 1, indicating exposures of concern.

## 2. Materials and Methods

### 2.1. Study Participants

Participants included subjects from NHANES cycles G-I (2011–2016) with human biomonitoring (HBM) values for the eight compounds, including phthalates (urinary): Mono-ethyl phthalate (MEP), Mono-n-butyl phthalate (MBP), Mono-benzyl phthalate (MBzP), Monoisononyl phthalate (MINP), and bisphenol A, (BPA); as well as perfluorinated compounds (serum): Perfluorohexane sulfonic acid (PFHxS), Perfluorononanoic acid (PFNA), and Perfluorooctane sulfonic acid (PFOS).

Demographic values included age, sex, pregnancy survey question, and NHANES respondent identifiers (i.e., survey sampling weights and sequence number). Survey sampling weights were included as a measure to evaluate how representative the collected dataset is to the population. For further analysis, female participants between 15 and 46 years of age were selected and represent women of reproductive age. The list of all analyzed variables retrieved from the NHANES database is included in Appendix A.

### 2.2. Regulatory Values

The International Society of Exposure Science (ISES) has created the international HBM Working Group (i-HBM) and developed a Biomonitoring Guidance Value (HB2GV) Dashboard which contains a compilation of currently available human biomonitoring guidance values developed by international organizations (https://biomonitoring.shinyapps.io/guidance/ (accessed on 25 January 2023)). The purpose of the tool is to facilitate the search for guidance values developed for specific chemicals. The tool contains Reference Dose (RfD) values from in vivo studies, Biomonitoring Equivalent (BE) levels, and values derived on the basis of toxicological and epidemiological studies that suggest no risk (HBM-I) or serve as thresholds to initiate action (HBM-II). For ease of general discussion, we subsequently refer to these as BE values. These values provide the opportunity to compare human biomonitoring data from human studies to regulatory guidance values. Such published BE values are tabularly presented for comparison to measured concentrations in NHANES (see Section 3, Table 2), and for comparison to results from the mixture approach of SMACH.

### 2.3. From In Vitro to Cohort

The reference mixture described in Caporale et al. [3] for use in SMACH consists of 8 EDCs with human-relevant mixing proportions given in Table 1. Derivation of mixing proportions and description of the tested in vitro assay have been described previously [3]. In brief, analysis of biomonitoring data collected during gestation from a Swedish pregnancy cohort, the SELMA study, highlighted 8 EDCs associated with language delay based on weights from a Weighted Quantile Sum (WQS) regression analysis (e.g., [3]). Biomonitoring data were derived from measurements in urine or plasma at the 10th week of gestation (GW10). Caporale et al. [3] proposed a method to determine the maternal plasma concentrations for all 8 EDCs, and the derived proportions were defined as MIX N (Table 1). MIX N was assessed using the XETA assay [6] in a dose response (0.1–1000X, with 1X concentrations presented in Table 1). The derived PoD of MIX N from the XETA assay was the basis to perform a SMACH analysis in the SELMA cohort. For those women with exposures sufficiently similar to MIX N, a SMRI was calculated and exposures of concern were determined based on values exceeding 1.

Input data for SMACH analysis are plasma concentrations in μM. For PFHxS, PFNA, PFOS, NHANES measurements were performed in serum and expressed in μg/L. For MEP, MBP, MBzP, MINP, and BPA, NHANES measurements were performed in urine (ng/mL) and toxicokinetic modeling was performed to estimate plasma concentrations. To do that, the solve_pbtk function of *httk* was used [7,8]. The *httk* is an R software package for high-throughput toxicokinetics that integrates (a) calculation of chemical parameters to describe partition across modeled tissues based on pKa, logKow, molecular weight (MW) and fraction unbound, (b) experimentally derived hepatic clearance rates for more than 9600 chemicals, and (c) physiological parametrization for toxicokinetic simulations [7]. Chemical parameters were available via *httk* for BPA, MBP and MBzP. To parametrize for MEP and MINP, physico-chemical parameters were calculated using the Marvin Protonation plugin (MW: 194.18; 292.4, logKow: 1.99; 5.02, pKa: 3.08; 3.08) [9]. These phthalates are the primary metabolite of their respective parent compound, diethyl phthalate (DEP) and diisononyl phthalate (DINP), for which *httk* parametrization is available. It was assumed that MEP and MINP had the same fraction unbound and intrinsic hepatic clearance as their parent compounds.

To derive conversion factors of studied chemicals between urine and plasma default physiological parameters were used for solve_pbtk function of *httk* for BW = 78.5 kg. Modeling was performed for all chemicals administered 3 times a day and daily dose 1 mg/kg BW. Estimates for plasma and urine concentrations were recorded for day 98. For toxicokinetic modeling using *httk*, a linear relationship is assumed between dose and predicted concentrations [10]. Per model compound, the following procedure was applied.

i.Expected urine concentration (C*_urine–httk_*, μg/L) using the following formula:(1)$$C_{urine–httk}\;\left(\mu g/L\right)=\frac{A_{urine}\left(\mu mol\right)*MW}{urine\;volume\;per\;day\;\left(L\right)}$$
where *A_urine_* is the amount of renally excreted compound per day (*μmol*), *MW* is molecular weight of simulated compound (Table 1), and urine volume per day is the expected daily urine volume for an adult woman adjusting for spot urine conversion (i.e., 1.6 L [11]).

ii.Urine to plasma conversion factor (CF*_httk_*) was calculated:$$CF_{httk}=C_{\begin{array}{c} plasma–httk \end{array}}\;\left(\mu M\right)\;/\;C_{urine–httk}\;\left(\mu g/L\right)\;$$
where *C_plasma–httk_* is the predicted plasma concentration in a steady state.

For all case chemicals measured in spot urine, analytical measurements include deconjugation [12,13], which means that all simulated compounds were the ones quantified in NHANES urine samples. Therefore, conversion factors were applied directly to NHANES measured urinary concentrations (ng/mL) of MEP, MBP, MBzP, MINP, and BPA to calculate plasma concentrations (μM) using the following formula:(2)$$C_{plasma–NHANES\;}\left(\mu M\right)=\left[NHANES\;measured\;urine\right]\;\left(ng/mL\right)*\;CF_{httk}$$
where *C_plasma–NHANES_* (*μM*) is the predicted plasma concentration after conversion.

### 2.4. Testing for Sufficient Similarity

Following Marshall et al. [14], we tested for sufficient similarity to the experimentally evaluated reference mixture (Table 1) using an equivalence testing approach. In short, we evaluated the similarity between an individual’s plasma concentrations to the reference mixture using the Euclidean distance between the benchmark dose (BMD) of the two mixtures. The BMD, and lower one-sided confidence limit BMDL, for the reference mixture is based on experimental results; in the data poor case described by Marshall et al. [14] (Equation (5)), we estimated the BMD for the ith subject’s mixture by adjusting the total dose of the mixture to that of the reference mixture (i.e., ${\hat{T}}_{r}$). The estimated distance measure and its variance estimate are then given by
$${\hat{d}}_{i}={\hat{T}}_{r}\sqrt{\overset{c}{\underset{j=1}{\sum }}{\left(a_{jr}-a_{ji}\right)}^{2}}\;\mathrm{and}\;\mathrm{var}({\hat{d}}_{i})=\mathrm{var}({\hat{T}}_{r})\left\{\overset{c}{\underset{j=1}{\sum }}{\left(a_{jr}-a_{ji}\right)}^{2}\right\}$$
where *a_jr_* is the proportion of the jth component in the reference mixture and *a_ji_* is the proportion of the jth component in the ith subject’s mixture, and
$$\overset{c}{\underset{j=1}{\sum }}a_{jr}=\overset{c}{\underset{j=1}{\sum }}a_{ji}=1$$

Using equivalence testing methodology, we tested for sufficient similarity using the principle of confidence interval inclusion, i.e., we claimed sufficient similarity in an alpha-level test when the upper confidence interval on *d* does not exceed the radius of the similarity region, R.

The similarity region assumed in Caporale et al. [3] was based on the benchmark response (BMR) from the XETA assay established by the OECD guideline test n° 248 of 12% reduction. The similarity region was assumed to be within an additional 12% reduction, i.e., ED_24%_–BMD_BMR = 12%_ = 0.89 (in units of 1X) on the log10(conc + 1) scale.

This test for sufficient similarity was conducted on all subjects in the selected NHANES data. NHANES uses a complex, multistage, probability sampling design where inference can be made that is representative of the civilian, non-institutionalized US population. Sampling weights were used to calculate summary statistics for the subset of NHANES who were women of reproductive age and were determined to be sufficiently similar to Mix N.

### 2.5. Calculating the Similar Mixture Risk Index

The BMD and BMDL were calculated using the experimental XETA study [3] in units of 1X and on the log10(conc + 1) scale. The BMD was estimated as 14X with BMDL = 8X (Appendix A from Caporale et al. [3]). To translate back to the concentration scale,
$$B\hat{M}D_{conc}=(10^{B\hat{M}D}-1)\sum_{j=1}^{c}Cp_{jr}\;\mathrm{and}\;B\hat{M}DL_{conc}=(10^{B\hat{M}DL}-1)\sum_{j=1}^{c}Cp_{jr}$$
where *Cp*_jr_ is the plasma concentration (mols/L) of the jth component in the reference mixture.

A SMRI was calculated for all women with sufficiently similar mixture composition to the reference mixture. That is, the estimated BMD from the experimental study was used to construct an index analogous to a hazard quotient:$$SMRI=\sum_{j=1}^{c}\frac{Cp_{j}(mols/L)}{BMDL_{j}(mols/L)}$$
where *BMDL_j_* is the concentration of the BMDL from the jth component in the mixture. Assuming the mixing proportions are equivalent between exposures for subjects considered to be sufficiently similar to the reference mixture (i.e., $a_{ij}=a_{rj},\;j=1,\ldots ,c)$, the *SMRI_ref_* index for the ith subject reduces to
$$SMRI_{ref}=\frac{T_{i}}{T_{r}}\overset{c}{\underset{j=1}{\sum }}\frac{a_{ij}}{a_{rj}}=\frac{T_{i}}{T_{r}}c$$
where *T* is total dose.

To illustrate in a simple example of 3 chemicals (e.g., A, B, and C), assume the concentrations for the ith subject are 2.52, 2.7, and 0.78 units, respectively, and the concentrations of the estimated BMDL are 3.2, 4.0, and 0.8 units. Thus, the total concentration of the 3 chemicals is *T_i_* = 6 for the ith subject and the BMDL is *T_r_* = 8 units. The mixing proportions for the reference mixture of the 3 chemicals are 0.4, 0.5, 0.1, and the mixing proportions for the ith subject are 0.42, 0.45, 0.13. Then, analogous to the hazard index (HI), $SMRI_{HI}=\left(\frac{2.52}{3.2}+\frac{2.7}{4}+\frac{0.78}{0.8}\right)=0.7875+0.675+0.975=2.4375$. Here, the exposures are all below the corresponding BMDL for each chemical from the reference mixture; however, their sum exceeds the value of 1. In comparison and more conservatively, we calculate the *SMRI_ref_* for the sufficiently similar subjects assuming the mixing proportions are set equal to the reference mixture, i.e., $SMRI_{ref}=\frac{6}{8}*\left(3\right)=2.25$. Summary statistics and a histogram of the $SMRI_{HI}$ values demonstrate the distribution of the index where values exceeding 1 indicate exposures of concern.

All calculations were conducted in R using the survey, dplyr, and haven packages and are available in the Appendix A.

## 3. Results

NHANES data were extracted from the CDC website from the 2011-16 cycles. There were *n* = 5735 participants with concentrations from the eight EDCs in MIX N. Using sampling weights, these participants represent nearly 242 million non-institutionalized people in the United States (Table 2). The concentrations are expected to be somewhat similar between the estimates for the full population and women of reproductive age (Table 2).

Additionally, included in Table 2 are published biomonitoring guidance values, retrieved by the HB2GV Dashboard (https://biomonitoring.shinyapps.io/guidance/ (accessed on 25 January 2023)). As seen in Table 2, the BE values generally exceed the 95th percentiles from NHANES, with the notable exception of PFOS.

In contrast to comparing human exposures to BE values one chemical at a time, we used a similar mixture approach. The first step is to determine the women of reproductive age who have sufficiently similar mixtures to the reference mixture (Table 1). In testing sufficient similarity per woman, chemical concentrations were expressed in μM of plasma, either estimated or measured. The urinary concentrations from the phthalates and BPA were translated to plasma estimates using conversion factors derived based on PBTK modeling in Table 3.

To illustrate the workflow, we take as starting point NHANES MBP measurement of 31 ng/mL in urine. The first step is toxicokinetic modeling to estimate MBP urine and plasma concentrations in a steady state. For a dosing regimen of 1 mg/kg BW/day, a 78.5 kg woman is estimated to have *C_plasma–httk_
*_(***MBP***)_ = 8.61 μM. For the same individual, the MBP renal excretion per day is estimated to be *A_tubules_
*_(***MBP***)_ = 1200 μmols, and the expected daily urine volume adjusting for spot urine conversion to be 1.6 L [11]. From that, we can calculate concentration in urine sample (*C_urine–httk_
*_(***MBP***)_, μg/L) with Equation (1):$$C_{urine–httk\;\left(MBP\right)}=\frac{\left[1200\;\mu mol\right]\left[222.24\frac{g}{mol}\right]\;}{\begin{array}{c} 1.6\;L \end{array}}=166680\;\frac{\mu g}{L}=166680\;\frac{ng}{mL}$$

In NHANES, measured concentrations are expressed in ng/mL, μg/L are equivalent units and can be used interchangeably. It is assumed that the simulated physiology is representative of the US population and that exposure dose is linearly correlated with urine excretion within the studied dose range [10]. Therefore, the ratio between *C_plasma–httk_
*_(***MBP***)_ (μM) and *C_urine–httk_* (*ng*/*mL*) can be used as a conversion factor for estimating plasma concentrations from measured urine concentrations per chemical. For our example of 31 ng/mL, the estimated plasma concentration (Equation (2)) would be
$$C_{plasma–NHANES\;\left(MBP\right)}=\left[31\frac{ng}{mL}\right]*\frac{8.61\;\mu M}{\begin{array}{c} 166680\frac{ng}{ml} \end{array}}=31\times 0.0000517=0.00160\;\mu M\;$$

Similar calculations were conducted for all subjects for the five compounds measured in urine. Measured levels of the three PFAS chemicals were transformed into μM units by scaling to corresponding MW (Table 1).

Following Marshall et al. [14], we tested for sufficient similarity to the experimentally evaluated reference mixture using an equivalence testing approach. In short, in the data-poor case [14], the distance between each subject’s mixing proportions and the reference mixture is calculated and translated to the scale of the estimated PoD of the reference mixture. Sufficient similarity is tested per subject by comparing the upper 95% confidence limit on the distance to the similarity region.

Using the R packages *svydesign*, *svymean*, and *svglm* for survey sampling estimation with survey sampling weights to infer to a nationally representative population, 66% of the women of reproductive age in these NHANES cycles were determined to have sufficiently similar mixtures to the reference mixture (i.e., nearly 38 million women). For those women with sufficiently similar mixtures to Mix N, we estimated the SMRI_HI_ (mean = 0.271; SE = 0.011) and SMRI_ref_ (mean = 0.139; SE = 0.004). The SMRI_HI_ uses the measured mixing proportions for each woman, while the SMRI_ref_ sets the proportions to the reference mixture. By comparing their concentrations to the PoD, none of the values of SMRI_ref_ exceeded 1, but 2.8% had SMRI_HI_ values exceeding 1, i.e., 1.1 million women were estimated to have SMRI_HI_ values exceeding 1.

Finally, an exploratory analysis revealed characteristics of women of reproductive age with higher SMRI_HI_ values in a regression model with survey sampling weights (Table 4). Older women were less likely to have high values (*p* = 0.021). Mexican American and other/multi race women were significantly more likely to have lower SMRI_HI_ values compared to non-Hispanic White women (*p* = 0.002, *p* = 0.019, respectively).

## 4. Discussion

Caporale et al. [3] integrated experimental and epidemiological evidence to establish mechanistic and correlative evidence for neurodevelopmental adversities of an EDC mixture associated with language delay, i.e., MIX N. The construction and evaluation of the mixture were based on the SELMA study. MIX N proportions were derived using a one-compartment model, based on urinary excretion factors for bisphenols and phthalates, and measured values of the PFAS chemicals in serum [3]. The mechanistic evidence highlighted herein was based on an in vivo model that confirmed thyroid hormone signaling as a key vulnerability to the experimentally evaluated mixture. The resulting dose–response relationship was used to estimate a point of departure to use in a risk assessment metric from a SMACH. Based on the results of the XETA studies, in the SELMA study 96% of the pregnant women were determined to have sufficiently similar exposures to the reference mixture. A SMRI was calculated from these women comparing their exposures to the PoD and 54% of all the women had a SMRI above 1.

Our objective in the current study was to follow the SMACH steps in another population in comparison to the PoD from MIX N. We considered women of reproductive age in the US NHANES (2011–2016 cycles) study. Urine to plasma ratios were estimated based on PBTK modeling to account for physiological differences. We determined that 66% of them had sufficiently similar exposures to MIX N, and 2.8% of the subset had SMRI_HI_ values exceeding 1, indicating a level of concern for 1.1 million women. Women most likely to have lower exposure levels were older women, and Mexican American and other/multi race women. The difference of occurrence of the SMRI > 1 between the two cohorts may be attributed to differences in exposure patterns between Sweden and the US in addition to the different modeling approaches to derive urine to plasma ratios and genetic variation that might affect toxicokinetic profiling.

Regulatory guidance values are derived from experimental studies. In particular, the RfD is an estimate of daily exposure to humans (including sensitive subgroups) that is likely to be without an appreciable risk of noncancer health effects during a lifetime. For many environmental chemicals, toxicokinetic modeling is available to translate RfDs to biomonitoring equivalent (BE) values [15]. Similarly, HBM-I values correspond to the concentration of a compound in human biological material below which no adverse health effects are expected to occur; in contrast, HBM-II values correspond to the concentration in human biological material which, when exceeded, may lead to health impairment which is considered as relevant to exposed individuals [16,17].

Epidemiological studies may provide evidence of associations between exposures and health effects, with more evidence from cohort studies compared to cross-sectional studies. Geographical and temporal considerations are key when cohort-to-cohort comparisons are performed since exposure patterns and chemical legislation may change over time and countries. Another key consideration is accounting for physiological and genetic variability between cohorts, and the extent that these factors would affect administration, distribution, metabolism, and excretion (ADME). On the other hand, dose–response studies in experimental settings provide more causal evidence linking exposures to adverse health effects. Therefore, crystallized epidemiological associations to interpretable in vitro evidence followed by in vitro to in vivo extrapolation protocols allow for the transfer of knowledge from one cohort to another, which transcend the limitations of a direct cohort-to-cohort comparison.

A similar mixture approach (SMACH) incorporates experimental evidence of adversities from a human-relevant mixture and compares its estimated PoD to biomonitoring concentrations. Epidemiological observations from one cohort are anchored with experimentally derived evidence. The derived PoD is relevant to plasma concentrations; therefore, by using toxicokinetics, it is possible for cross-study comparisons where geographical and temporal variables of the study are not to be considered as confounding factors. For the present work, toxicokinetic modeling was performed with the *httk* package developed by the US Environmental Protection Agency to translate urinary concentrations from study participants to estimate plasma concentrations so that all compounds are expressed in the same units as the mixture. PBK parametrization was possible with the exception of MINP and MEP, where hepatic clearance data were limited. In turn, it was assumed that hepatic clearance is the same between parent (i.e., DINP and DEP) with primary metabolite (i.e., MINP and MEP). We begin this translation herein by making simplifying assumptions about physiology (i.e., defined physiology representative of women of reproductive age in US, clearance in pregnant and non-pregnant population is the same). It is important to acknowledge the uncertainties that come from assumptions in parametrization of clearance for MINP and MEP, the impact of inter-individual and dosing regimens differences might have on estimates ADME and subsequently calculation of urine to plasma ratio for MIX N compounds. Another point of uncertainty that needs to be investigated further is the use of appropriate metabolites to monitor MINP. More specifically, MINP undergoes Phase I and Phase II metabolism, and it is assumed that only 2.1% of the orally administered DINP dose corresponds to MINP in urine. In turn, OH-MINP, a Phase I metabolite of MINP, corresponds to ~18–20% of the orally administered DINP dose, and it would potentially be a more suitable metabolite to monitor both DINP and MINP exposure [18]. The extension of these assumptions and how these uncertainties can be addressed are currently under exploration by our group.

We have provided the R code (see SI) used to test for sufficient similarity to MIX N and then calculate the SMRI indices from subjects determined to have sufficiently similar exposures. We are currently working with other international pregnancy cohorts to implement this code with their measured EDC exposures that are included in MIX N. Evidence to date demonstrates that pregnant women or women of reproductive age have exposures to EDCs in MIX N that may be harmful to the neurodevelopment of their children.

The work of Caporale et al. [3], as a paradigm, may serve as a guide to other researchers who may want to incorporate experimental evidence of the adverse effects of other human-relevant mixture(s) on vulnerable populations in international cohorts or biomonitoring studies. This integrative strategy emphasizes the need to consider mixtures during chemical testing and provides an approach that links experimental results with relevant human exposures.

These findings indicate that a reference mixture of chemicals identified in a Swedish cohort—and tested in an experimental model for the establishment of (PoDs)—is also of health relevance in a US population.

## Figures and Tables

**Table 1 toxics-11-00331-t001:** MIX N concentrations based on SELMA biomonitoring data, where neurodevelopmental adversity was observed [3].

Compound ^1^	CAS	Molecular Weight (g/mol)	Concentration (mol/L)	Mixing Proportions
MEP	2306-33-4	194.18	2.7 × 10^−8^	0.27
MBP	131-70-4	222.24	2.26 × 10^−8^	0.23
MBzP	2528-16-7	256.25	1.05 × 10^−8^	0.11
MINP	106610-61-1	292.4	2.06 × 10^−8^	0.21
BPA	80-05-7	228.29	4 × 10^−9^	0.04
PFHxS	355-46-4	400.12	3.2 × 10^−9^	0.03
PFNA	375-95-1	464.08	1.1 × 10^−9^	0.01
PFOS	1763-23-1	500.13	1.03 × 10^−8^	0.10

^1^ Full names in Section 2.1.

**Table 2 toxics-11-00331-t002:** Summary statistics (median, 95th percentile) for MIX N compounds for levels (μg/L) in urine (phthalates and BPA) and plasma (PFAS) as documented by NHANES (2011-16). Biomonitoring guidance values (μg/L) are provided for comparison. The minimum guidance value is displayed if more than one is given (https://biomonitoring.shinyapps.io/guidance/ (accessed on 25 January 2023)).

Compound	Biomonitoring Equivalent (BE) Values	NHANES *n* = 241,771,564 ^1^		NHANES Women (15–46 Years Old) *n* = 57,748,243 ^1^	
		Median	95th Percentile	Median	95th Percentile
MEP	18,000 (age 6 and up)	30.3	531.4	31.0	504.8
MBP	200 (age 6 and up)	9.5	46.7	10.0	50.6
MBzP	3800 (age 6 and up)	4.1	32.5	4.8	41.7
MINP	390 (age 6 and up)	0.6	12.5	0.6	17.5
BPA	100 (children) 200 (adults)	1.2	7.8	1.2	7.3
PFHxS	-	1.3	5.4	0.7	3.2
PFNA	-	0.7	2.2	0.5	1.8
PFOS	5	5.5	19.4	3.3	10.4

^1^ With sampling weights.

**Table 3 toxics-11-00331-t003:** Predicted urine and plasma concentrations (μM) of Mix N compounds for daily dose 1 mg/kg BW, and their ratios (i.e., CF*_httk_*) to be used as conversion factors for urine to plasma concentrations for NHANES-derived measurements.

Compound	Predicted Concentrations–*httk*		CF*_httk_* Samples Measured in:	
	C_plasma_ (uM)	C_urine_ (ng/mL)	Urine (ng/mL) to Plasma (μM)	Plasma (μg/L) to (μM)
MEP	0.018	2670	0.0000068	1
MBP	8.614	166680	0.0000517	1
MBzP	44.39	928906	0.0000478	1
MINP	0.013	1261	0.0000099	1
BPA	0.949	24256	0.0000391	1
PFHxS			1	0.00249
PFNA			1	0.00215
PFOS			1	0.00199

**Table 4 toxics-11-00331-t004:** Parameter estimates and *p* values from a linear regression model (using survey sampling weights) of log(SMRI_HI_) in the women determined to be sufficiently similar to the reference mixture.

Coefficient	Estimate	Standard Error	*p* Value
Intercept	−1.261	0.144	<0.001
Age (years)	−0.010	0.004	0.021
Poverty Index ^1^	−0.025	0.077	0.747
Race ^2^: MexAmer	−0.267	0.080	0.002
Race: OtherHisp	0.071	0.093	0.450
Race: NonHispBlack	0.884	0.069	0.230
Race: Other/Multi	−0.244	0.010	0.019

^1^ log(poverty index + 1); ^2^ Non-Hispanic White is the reference group.

## Data Availability

The NHANES data are publicly available (https://wwwn.cdc.gov/nchs/nhanes/).

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
