# Peer review of "From Cohort to Cohort: A Similar Mixture Approach (SMACH) to Evaluate Exposures to a Mixture Leading to Thyroid-Mediated Neurodevelopmental Effects Using NHANES Data"

_toxics, 2023, doi:10.3390/toxics11040331_

Round 1

Reviewer 1 Report

General comments

I appreciate that the authors are building on previous work and do not want to go into too much detail on restating the methods from Caporale et al. However, I am not entirely clear on the use of XETA assay results to serve as the basis for determining similarity. Could the authors please provide a little more explanation?

Please review all equations in the manuscript. Only equation 1 and 2 are numbered. I think labeling equations consistently would be helpful. Additionally, all equations in the “Testing for sufficient similarity” section are missing symbols and have empty boxes. I am assuming this is a conversion issue when the pdf was made – perhaps the program used to make the equations was not compatible, but not sure.

Perhaps the authors could compare their findings to other work that has looked at similar chemical classes in populations – are the same trends noted (lower levels of these chemicals in older and Mexican, Other/Multi women)? Any speculation on what is behind these differences?

Specific comments

Line 106: Could you specify whether each chemical was associated with language delay or whether they were associated using a mixture model?

Line 113: Should it be “at-risk populations”? How did the SMRI calculations confirm that at risk populations were appropriately identified?

Line 115: Change to “based on” and “a subset cohort”.

Line 136: Delete extra period.

Equation 1: Make urine a subscript to “A” as in “Aurine

Line 164: Not sure why “for” is italicized.

Line 186: I am not sure what is meant by “Sampling weights were used to calculate summary statistics…” Could more explanation be provided?

Line 204: Make “ref” a subscript of SMRI in the equation.

Line 206: I have not heard the term “toy example”, is it specific to statistics? I wonder if “simple example” would be more universal?

Line 212: Add a period after 2.4375.

Line 229: Does BE refer to the Biomonitoring guidance values in Table 2? Suggest using consistent terminology.

Table 3: Check that urine and plasma are subscripts to “C”.

Line 244: Should there be a dash between mg and kg to make it 1 mg/kg/day?

Line 248: There is a box at the end of the sentence and no period.  

Line 275: Can you provide a bit more context to “using sampling weights”?

Line 296: What about plasma concentrations of PFAS? Didn’t they also contribute to the MIX N proportions?

Author Response

RESPONSE to REVIEWERS

Reviewer 1

We appreciate the reviewer’s thorough and thoughtful review.  It has indeed improved the paper. Specific responses to each point follow.

General comments

I appreciate that the authors are building on previous work and do not want to go into too much detail on restating the methods from Caporale et al. However, I am not entirely clear on the use of XETA assay results to serve as the basis for determining similarity. Could the authors please provide a little more explanation?

Response: We added a sentence linking the important role of the thyroid in brain development in lines 56-58.

Please review all equations in the manuscript. Only equation 1 and 2 are numbered. I think labeling equations consistently would be helpful. Additionally, all equations in the “Testing for sufficient similarity” section are missing symbols and have empty boxes. I am assuming this is a conversion issue when the pdf was made – perhaps the program used to make the equations was not compatible, but not sure.

Response: The equations look fine on the .doc version.  Perhaps the problem was in the pdf.

Perhaps the authors could compare their findings to other work that has looked at similar chemical classes in populations – are the same trends noted (lower levels of these chemicals in older and Mexican, Other/Multi women)? Any speculation on what is behind these differences?

Response: As stated, this regression analysis was exploratory mainly to demonstrate the approach of using the SMRI in a model.  Although we appreciate the reviewer’s suggestion to compare our findings to the literature, we respectfully consider further speculation is beyond the scope of this paper.

Specific comments

Line 106: Could you specify whether each chemical was associated with language delay or whether they were associated using a mixture model?

Response: We added the phrase “selected based on WQS regression weights”

Line 113: Should it be “at-risk populations”? How did the SMRI calculations confirm that at risk populations were appropriately identified?

Response:  We rewrote the sentence to be more specific about what we calculated and in so doing removed the phrase.

Line 115: Change to “based on” and “a subset cohort”.

Response: We made the first edit, but deleted the phrase about a subset cohort

Line 136: Delete extra period.

Response: We made the change.

Equation 1: Make urine a subscript to “A” as in “Aurine

Response: We made the change.

Line 164: Not sure why “for” is italicized.

Response: We emphasized “for” to indicate the direction change from the usual hypothesis testing framework.  But since it is not clear, we removed the italics.

Line 186: I am not sure what is meant by “Sampling weights were used to calculate summary statistics…” Could more explanation be provided?

Response: We added a sentence about the complex sampling design such that with sampling weights inference is made on a representative population.

Line 204: Make “ref” a subscript of SMRI in the equation.

Response: We made the edit

Line 206: I have not heard the term “toy example”, is it specific to statistics? I wonder if “simple example” would be more universal?

Response: As the reviewer suggested, we changed to ‘simple example’

Line 212: Add a period after 2.4375.

Response: Thank you for your thorough review. We made the edit.

Line 229: Does BE refer to the Biomonitoring guidance values in Table 2? Suggest using consistent terminology.

Response: We made the edit.

Table 3: Check that urine and plasma are subscripts to “C”.

Response: We made the edit.

Line 244: Should there be a dash between mg and kg to make it 1 mg/kg/day?

Response: We made the edit.

Line 248: There is a box at the end of the sentence and no period.  

Response: We added a colon and a period at the end of the equation.

Line 275: Can you provide a bit more context to “using sampling weights”?

Response: We provided the R packages used to do the estimation with survey sampling weights

Line 296: What about plasma concentrations of PFAS? Didn’t they also contribute to the MIX N proportions?

Response: We added a phrase about the PFAS compounds which were measured in serum and did not need to be estimated with HTTK modeling.

Reviewer 2 Report

The study described in this article conducted a risk assessment for a mixture of 8 chemicals (4 phthalates, BPA, 3 PFAS) in US women of reproductive age using a point of departure for a similar mixture derived via a QVIVE approach based on the effect of disruption of thyroid hormone signaling in xenopus. The authors' methods for toxicokinetic modeling and for determining similarity between the reference mixture and mixtures of the same chemicals in the experimental cohort seem sound. However, this reviewer questions the inclusion of PFAS chemicals in this mixture study. PFAS chemicals have not consistently been associated with thyroid disease, effects on thyroid hormones, or adverse neurodevelopmental effects in humans (see EFSA, 2018, 2020). The authors' discussion section should discuss in more detail the scientific evidence for PFAS effects on thyroid function in humans.

Author Response

Reviewer 2

The study described in this article conducted a risk assessment for a mixture of 8 chemicals (4 phthalates, BPA, 3 PFAS) in US women of reproductive age using a point of departure for a similar mixture derived via a QVIVE approach based on the effect of disruption of thyroid hormone signaling in xenopus. The authors' methods for toxicokinetic modeling and for determining similarity between the reference mixture and mixtures of the same chemicals in the experimental cohort seem sound. However, this reviewer questions the inclusion of PFAS chemicals in this mixture study. PFAS chemicals have not consistently been associated with thyroid disease, effects on thyroid hormones, or adverse neurodevelopmental effects in humans (see EFSA, 2018, 2020). The authors' discussion section should discuss in more detail the scientific evidence for PFAS effects on thyroid function in humans.

Response: The reviewer’s point is an interesting one. The study of PFAS chemicals is indeed complicated in human studies and there are inconsistent results.  However, the reviewer may be interested in a recent paper relating PFAS and thyroid hormones in the SELMA study (Derakhshan et al, 2022; DOI: 10.1016/j.envint.2022.107420). From that study, there is evidence of a complex correlation pattern among the PFAS in the SELMA pregnant women, and all three of the PFAS included in Mix N have significant associations with one or more thyroid hormones in single chemical analyses. Similar evidence is brought forward from other cohort studies (e.g. Yao et al, 2022 DOI: https://doi.org/10.1016/j.envres.2021.112561) and reviews (e.g. Gundacker et al, 2022 DOI: https://doi.org/10.3390/toxics10110684).